# Time-Dependent Model for Brittle Rocks Considering the Long-Term Strength Determined from Lab Data

James Innocente [1], Chrysothemis Paraskevopoulou [2,*] and Mark S. Diederichs [1]

[1] Department of Geological Sciences and Geological Engineering, Queen's University, Kingston, ON K7L 3N6, Canada; j.innocente@queensu.ca (J.I.); diederim@queensu.ca (M.S.D.)
[2] School of Earth and Environment, University of Leeds, Leeds LS2 9JT, UK
* Correspondence: c.paraskevopoulou@leeds.ac.uk

**Abstract:** The excavation of tunnels in brittle rocks with high in-situ strengths under large deviatoric stresses has been shown to exhibit brittle failure at the periphery of tunnels parallel to the maximum in-situ stress. This failure can either occur instantaneously or after several hours due to the strength degradation that is implicitly and indirectly considered in typical brittle constitutive models. While these models are powerful tools for engineering analyses, they cannot predict the time at which brittle rupture occurs, but rather, they show a possible failure pattern occurring instantaneously. In this paper, a model referred to as the long-term strength (LTS) model is introduced and implemented into FLAC2D. The model is built as a modified version of the CVISC model, introduced by Itasca, by adding a strength decay function. This function is developed from lab-scale time-to-failure (TTF) data. The LTS model is verified against its corresponding analytical solution using a constant stress creep lab test and implemented into a tunnel-scale model using the geometry, stress, and geologic conditions from the Atomic Energy of Canada Limited Underground Research Laboratory (AECL URL). The results of the LTS tunnel model are then compared to an identical model using the Cohesion Weakening Friction Strengthening (CWFS) approach.

**Keywords:** long-term strength and time-to-failure; time-dependency and creep; non-Newtonian viscosity





## 1. Introduction

The ongoing development and calibration of models for rock strength and deformation around tunnel peripheries is important in rock mechanics. The need for constitutive models in the field of rock mechanics allows engineers and scientists to obtain estimates of yield zones, displacements, and ground settlements so that support requirements and excavation methodologies can be optimized [1,2]. Classical rock strength models, such as the Mohr–Coulomb criterion [3] and the Hoek–Brown criterion [4–6] have been developed for use in moderately-jointed rock masses where failure is the result of block rotation, or where 30 < GSI < 65 [7]. In more massive rock masses under low-to-moderate confinement, failure becomes the result of extensile processes rather than shear based [8,9] which the classical failure criteria do not consider.

Early attempts to capture brittle behaviour in rock masses have used an iterative elastic approach as shown in [10,11]. It has been shown that when the intact strength of rock masses initially has a near-zero frictional strength, the failure behaviour at the tunnel periphery matches those observed in the AECL URL [11]. The mechanics of near-zero frictional strength show that when intact, the strength that is controlled by cohesion and friction is only mobilized when rupture occurs. Following this work, the CWFS approach was introduced by [12] and the DISL approach by [8].

The strength of brittle rocks and rock masses is also considered time-dependent, e.g., [8,13–18] where brittle failure at the excavation scale can manifest immediately or after

some time due to subdued crack growth and interaction. The lower limit at which these cracks can develop and grow with time is the crack-initiation stress threshold [8]. The authors of [19] compiled several long-term strength tests conducted on brittle rocks from published sources and showed that as the time-to-failure increases, the lower limit of stress at which the rocks fail approaches the crack-initiation threshold.

The CWFS and DISL methods have been shown to capture the brittle behaviour of rocks at the excavation scale but fail to capture the time-to-failure and time-dependent deformations in these models which can be an important aspect to consider in an engineering design. An alternative approach to modelling brittle failure with time-dependent deformations based on time-to-failure lab tests is proposed and explored, and the subsequent results are validated against an equivalent CWFS analysis. The primary aim of this paper is to develop a time-dependent model, the long-term strength (LTS) model, that can adequately capture creep and strength degradation leading to progressive failure and rupture in brittle rocks. This paper provides an overview of the existing CWFS model and its applicability when modelling underground excavations in brittle rock. The CWFS model is used as a verification tool for the proposed LTS model to show that both the magnitude and the geometry of failure around a circular tunnel in brittle rock are equivalent.

## 2. Background

This section provides an overview of the different types of time-dependent behaviours observed at the lab or at the excavation scale as well as the failure modes and mechanisms of brittle rocks around and away from tunnel peripheries.

### 2.1. Failure in Brittle Rocks

The proper classification of rock masses and their associated in-situ strength at the excavation scale has been a significant research topic for many researchers, e.g., [4,5,7,8,20–28]. The use of engineering design equations, such as the GSI system, to determine rock strength parameters were developed and calibrated for use in moderately blocky rock masses (30 < GSI < 65), wherein the failure process is associated to block rotation. The author of [7] demonstrated that for more intact rocks (GSI > 65), the GSI equations are not valid because blocks cannot form without failure through intact rock first. Such failure typically occurs due to the formation of axial cracks parallel to the direction of the maximum applied stress, which is related to the tensile strength of the rock [7,29].

In recognition that classical failure criteria do not consider the effect of axial splitting, but rather shear fracturing, other approaches to modelling brittle failure, namely in continuum models, have been developed. Such models include the cohesion weakening, the friction strengthening (CWFS) model [11], and the damage initiation spalling limit (DISL) model [7]. These models are based on the respective crack-initiation (CI) and crack-damage (CD) thresholds as defined by [25]. When in-situ stresses are above the CI threshold, new fractures initiate and propagate with time, ultimately controlling the long-term strength (LTS) of the rock in low-to-moderate confinement. At higher confinement levels, the initiation of fractures becomes inhibited, changing the failure modes from tensile to shear rupturing [30,31], as shown in Figure 1.

At low-to-moderate confinements, the failure behaviour as governed by the DISL and CWFS approaches correspond well with empirical observations made in massive, brittle rock with stresses around the periphery of excavations at or exceeding the CI threshold, as performed in the following studies [7,8,32,33], among which a review of the CI and CD thresholds in various rocks found that the average CI to UCS ratio is between 0.4 and 0.55 whereas the average CD to UCS ratio is between 0.75 and 0.9. When rocks in low confinement are loaded to the CI threshold, they will experience continued fracture growth with time, leading to failure, whereas if they are loaded to the CD threshold, they will experience rapid crack growth and interaction, leading to sudden rupture (spalling), e.g., [22,34–36].

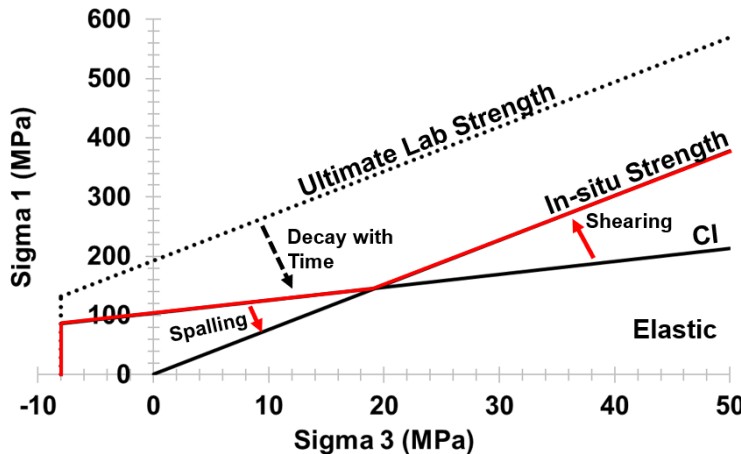

**Figure 1.** Strength envelope of the LdB granite at the AECL URL (CWFS) used to describe in−situ strength and development of brittle failure. Modified from [8,25].

The author of [8] provides an outline of which conditions using the DISL and CWFS approaches are most appropriate based on the ratio of UCS to tensile strength and geological strength index (GSI) from [24] shown in Table 1. The author of [37] later provides a set of guidelines for determining equivalent CWFS parameters based on the commonly used Mohr–Coulomb criterion. Using the strength data for the LdB granite at the AECL URL provided in [37] modified from [8], a typical CWFS analysis can be used as a baseline for comparison to the LTS model.

**Table 1.** Constitutive model selection based on rock strength and GSI, modified from [8]. Note that UCS and T indicate intact unconfined compressive strength and tensile strength, respectively.

| Strength Ratio | GSI < 55 | GSI = 55–65 | GSI = 65–80 | GSI > 80 |
|---|---|---|---|---|
| UCS/T < 9 | GSI | GSI | GSI | GSI |
| UCS/T = 9–15 | GSI | GSI | GSI | GSI or CWFS * |
| UCS/T = 15–20 | GSI | GSI or CWFS * | CWFS or GSI * | CWFS |
| UCS/T > 20 | GSI | GSI or CWFS * | CWFS | CWFS |

* indicates most appropriate analysis first.

### 2.2. Creep in Rock Mechanics

Creep in rock mechanics is defined as the accumulation of shear strain under constant stress without change in volume. Creep behaviour in solids has long been studied since the early 20th century by many researchers, e.g., [13,17,38–49]. Among the early experimental studies on creep [39,50] were performed on steel while [13] was the first researcher to apply the study of creep on geomaterials, namely on talc, shale, and crystals of halite and calcite at various levels of confinement. In [38], the author introduced the idea of three distinct stages of creep while studying creep in metals, as shown in Figure 2.

When a load is applied to a creeping material, its instantaneous deviatoric and volumetric behaviour is described by Hooke's Law [51]. After time is considered, the accumulated deviatoric strains increase at a decreasing rate (primary stage creep). If the load is held constant thereafter, the deviatoric strains increase but at a constant rate (secondary stage creep), after which the material may or may not enter the tertiary stage (yield). In theory, the volumetric strains do not change throughout the creep process; however, this assumption is only valid if the deviatoric strains develop from true creep processes such as solid diffusion, dislocation creep, or solution transfer, e.g., [13,41,52,53]. In practice, strain accumulation due to creep processes are only observed at the relatively short timescale in rock salt, potash, steel, and other ductile materials, e.g., [39,43,50]; however, true creep processes may also be observed in strong, brittle rocks given appropriate environmental conditions and longer time spans.

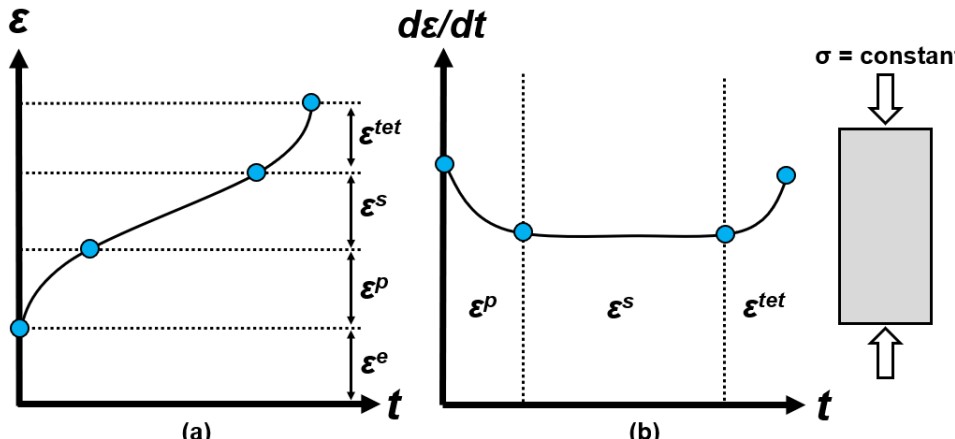

**Figure 2.** General creep curve in axial strain-time space for a specimen of rock under an applied constant load showing (**a**) the three stages of creep (primary, secondary, and tertiary) and (**b**) associated strain-rate time curve. The strain-rate curve is also referred to as the "bathtub" curve. Note that superscripts *e, p, s,* and *tet* denote elastic, primary, secondary, and tertiary, respectively.

At the tunnel scale, creep is often observed as the decrease in tunnel radius with time, e.g., [49,54]. This behaviour can also be associated with squeezing and swelling, which are attributed to weak and soft rocks [55,56]. Squeezing is defined as the advance of rock into a tunnel without a perceptible volume change due to the presence of micaceous minerals with low swelling capacity. Swelling is defined as the expansion of rock limited to rocks which contain clay minerals such as montmorillonite or other minerals with high swelling capacity [57].

Rheological Models

The term "rheology" refers to the branch of study related to the flow of liquids and solids under an applied force in which they deform plastically rather than elastically [58]. Rheological creep models are models that have been built up from simple mechanical analogues such as springs (Hookean elements), dashpots (Newtonian elements), and plastic sliders (St. Venant elements) as described by some constitutive model, typically the Mohr–Coulomb criterion. These elements can then be combined in series or parallel in many ways to describe the strain-time behaviour of rocks at the lab scale or in-situ. The author of [59] provides various examples of ways in which these elements can be arranged to describe different observed behaviours. The Hookean element is described by Hooke's Law [51] in which the displacement of a spring is linearly proportional to the stress acting on the spring and the stiffness of the spring.

The time-dependent aspect of rock deformation in rheology is represented by the Newtonian element which follows Newton's law of viscosity [60]. Newton's law of viscosity states that a material or fluid under applied constant stress will exhibit a constant rate of deformation with time. The material's resistance to this deformation is referred to as its viscosity, which is a material property and does not change with stress, time, or accumulated deformation. Conversely, non-Newtonian fluids are materials that do not obey Newton's law of viscosity because their respective viscosity is not constant at either given stress, strain-rate, or deformation level. One such material is Bingham plastic which behaves as a solid at low stress (does not flow) but flows as a viscous fluid at high stresses [61]. The Newtonian element can adequately capture the time-dependent aspect of rock deformation; however, it cannot capture the instantaneous response like the Hookean element. In addition, the mechanics of the viscous element allow strains accumulated with time to be fully recovered with reversal in boundary conditions regardless of the magnitude of accumulated strains, meaning the element lacks plasticity as well. A rheological model is a model that incorporates the Hookean, Newtonian, and St. Venant elements in some

combination. These models can be visco-elastic, visco-elastic-plastic, or elasto-visco-plastic. Table 2 highlights some common rheological models used in the study and the modelling of creeping materials, e.g., [49,59,62–64].

**Table 2.** Visco-elastic-plastic rheological models with their associated mechanical analogues; analytical solutions; and stress, strain–time behaviour, modified from [18]. Note that G, K, $G_K$, $\eta_K$, and $\eta_M$ denote the shear modulus, bulk modulus, Kelvin viscosity, and Maxwell viscosity, respectively, and p, q, t, and ε denote mean stress, deviator stress, time, and strain, respectively.

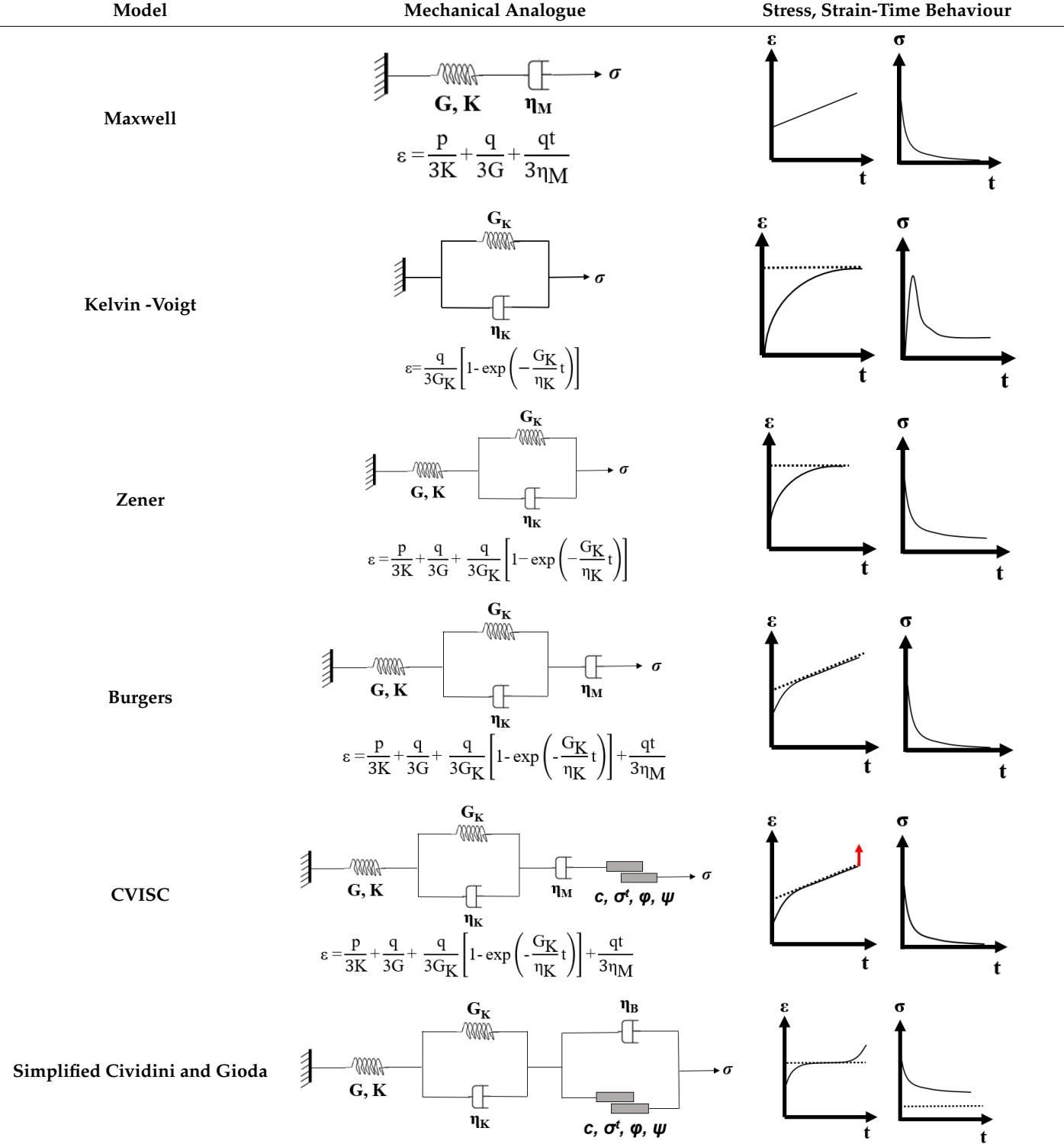

| Model | Mechanical Analogue | Stress, Strain-Time Behaviour |
|---|---|---|
| Maxwell | $\varepsilon = \dfrac{p}{3K} + \dfrac{q}{3G} + \dfrac{qt}{3\eta_M}$ | |
| Kelvin -Voigt | $\varepsilon = \dfrac{q}{3G_K}\left[1 - \exp\left(-\dfrac{G_K}{\eta_K}t\right)\right]$ | |
| Zener | $\varepsilon = \dfrac{p}{3K} + \dfrac{q}{3G} + \dfrac{q}{3G_K}\left[1 - \exp\left(-\dfrac{G_K}{\eta_K}t\right)\right]$ | |
| Burgers | $\varepsilon = \dfrac{p}{3K} + \dfrac{q}{3G} + \dfrac{q}{3G_K}\left[1 - \exp\left(-\dfrac{G_K}{\eta_K}t\right)\right] + \dfrac{qt}{3\eta_M}$ | |
| CVISC | $\varepsilon = \dfrac{p}{3K} + \dfrac{q}{3G} + \dfrac{q}{3G_K}\left[1 - \exp\left(-\dfrac{G_K}{\eta_K}t\right)\right] + \dfrac{qt}{3\eta_M}$ | |
| Simplified Cividini and Gioda | | |

The Maxwell, Kelvin–Voigt, Zener, and Burgers models are referred to as visco-elastic models. The CVISC model is a visco-elastic-plastic model introduced by [65] and the simplified Cividini and Gioda is an elastic-visco-plastic model introduced by [66]. It is important to note the distinction between 'elasto-plastic' and 'visco-plastic', denoting no connection between creep and plasticity and a direct connection, respectively. The connection between creep and plasticity is complex and often difficult to determine. The Burgers and CVISC models are preferable for practical applications [67]; however, there are limitations to the model as described in [68–70] and in this paper.

Other models to examine creep exist, such as empirical or phenomenological models, as well as general theories [64]. Empirical models are models built purely from curve-fitting of lab data from constant-stress or stress-relaxation tests and are generally given as closed form or differential solutions. General theories are the most advanced aspects of numerical modelling and are generally very robust in their use case. Perzyna's overstress theory is one such example of a general theory [71]. The further analysis and application of empirical models and general theories is out of the scope of this paper.

### 2.3. Interpreting Time-to-Failure Lab Results in Brittle Rocks

The most common method for determining the long-term strength of brittle rocks is by conducting a series of uniaxial compressive strength (UCS) tests to determine the average strength of the rock. This suite of tests should follow the methodology as outlined in [72–74]. The long-term strength tests comprise loading a standard cylinder of core to some stress that is less than its UCS but more than its crack initiation (CI) threshold. Once the desired stress is reached, it is held and the time to failure (TTF) is recorded and compared to the applied driving stress ratio (**DSR**) which is historically presented in Equation (1) as:

$$\mathbf{DSR}_{\sigma_3=0} = \frac{\sigma_1}{\mathbf{UCS}} \tag{1}$$

where:

- $\sigma_1$ is the applied axial stress.

Ref. [19] reviews the current state of practice for analysing the long-term strength of brittle rocks and proposes that the model shown in Figure 3 be used to calculate the TTF for igneous rocks when under uniaxial stress conditions. Based on an earlier formulation from [75], The author of [19] developed a set of Equations (2) and (3)

$$t_f = \left( -\frac{C - \ln(100\mathrm{DSR})}{A} \right)^{-\frac{1}{B}} \text{ for } t_f > 10 \text{ s and DSR} > \frac{\exp(C)}{100} \tag{2}$$

$$C = \ln\left( \frac{CI}{UCS} \times 100\% \right) \tag{3}$$

where:

- $t_f$ is the time-to-failure;
- C is an asymptote control parameter;
- A and B are curve-fitting constants that are determined empirically.

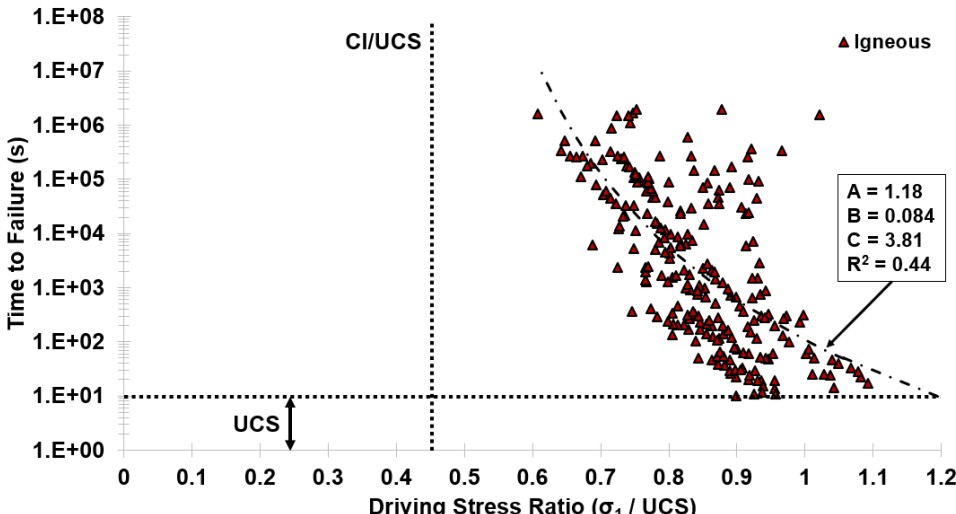

**Figure 3.** Long-term strength data for various rock types and their respective exponential model fits as presented in [19]. The average CI threshold [76] is added to represent the lower bound of long-term strength.

Note that time-to-failure is a function of intact material properties and does not change with time; therefore, it can be considered as a material property that is a function of applied stress. To be able to use the TTF equations as shown in Equations (2) and (3), the effect of confinement on strength must be considered; however, very few long-term strength tests have been conducted under confined conditions apart from the suite of tests conducted by [77,78] in which it is shown that the effect of confinement clearly affects the absolute TTF under constant applied stress. The TTF as shown in Equations (2) and (3) is a function of unconfined conditions only. To account for this, the DSR must be modified as **DSR\*** shown by [79] in Equation (4):

$$\mathbf{DSR}^* = \frac{\sigma_1 - \sigma_3}{\sigma_1^{\mathbf{p}} - \sigma_3} = \frac{\mathbf{q}}{\sigma_1^{\mathbf{p}} - \sigma_3} \tag{4}$$

where:

- $\sigma_3$ is the confinement, or minimum principal stress;
- $\sigma_1^{\mathbf{p}}$ is the peak strength of the rock at a given confinement level;
- **q** is the deviator stress.
- **DSR\*** is the modified DSR

The justification for the modified DSR equation is shown in Figure 4. In unconfined conditions, the DSR equation simplifies to that as shown in Equation (1). It is assumed that the same DSR under both unconfined and confined conditions will lead to the same TTF as shown in Equation (2). Equation (3) does not need further modification as the ratio of CI to UCS is also assumed constant with relatively small increases in confinement; however, this assumption is only valid within the spalling limit of the material, which is within the range of $10 \leq \sigma_1/\sigma_3 \leq 20$ [80,81]. It should also be noted that the DSR and TTF formulations are for two-dimensional problems, but they can be modified for use in three-dimensional problems.

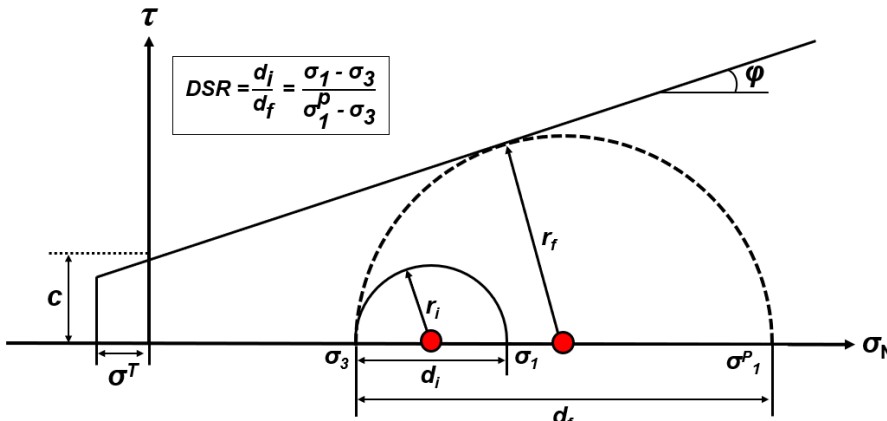

**Figure 4.** Schematic of variables used within the DSR equation for confined conditions using the Mohr−Coulomb criterion, modified from [79].

### 3. Proposed Long-Term Strength (LTS) Model

This section describes the long-term strength (LTS) model that explicitly considers time in two dimensions. The proposed LTS model is built on the existing CVISC creep model by modifying the attached Mohr–Coulomb plastic slider, as shown in Table 2. The modification includes degrading the strength of the criterion as per the TTF equations in Section 2.3. The process for degrading strength with time is outlined in this Section.

In two dimensions, the Mohr–Coulomb failure criterion is expressed in Equations (5) and (6) as:

$$\sigma_1^P = UCS + s\sigma_3 \tag{5}$$

$$s = \frac{1 + \sin \varphi}{1 - \sin \varphi} \tag{6}$$

where:

- $\varphi$ is the friction angle as shown in Figure 4.

Using Equation (5), the strength of the rock at the excavation scale can be determined throughout the FLAC2D [65] grid. With the data from Figure 3, the theoretical time-to-failure can be determined throughout the FLAC2D grid using the equations described below. A key assumption in the development of this model is that the failure resulting from stress corrosion is the result of a linear loss in cohesion with time that is a function of the in-situ stress conditions. This assumption is made based on the log-linear relation that laboratory data shows when brittle rock materials are subjected to constant load as [19] thoroughly analysed and discussed lab datasets from a range of rock materials. In the finite difference model, this can be achieved by introducing a damage variable (R), which is calculated as shown in Equation (7).

$$\frac{dR}{dt} = (1 - DSR)\left(\frac{1}{t_f}\right) \tag{7}$$

The damage is then iterated through time using the creep plugin available for FLAC2D. It should be noted that damage can be calculated directly with time; however, this limits the applicability of the equation to monotonic loading conditions only, whereas iterating the variable allows it to apply to more complex loading conditions such as those encountered in underground mines and other excavations. The damage function is then used to decrease the ultimate strength ($\sigma_1^P$) of each time-step and tensile strength ($\sigma^T$), as shown in Equations (8) and (10). The damage to ultimate strength is used to calculate the new cohesion, as per Equation (9).

$$UCS^* = R\sigma_1^P - s\sigma_3 \tag{8}$$

$$c^* = \frac{\text{UCS}^*(1 - \sin(\varphi))}{2\cos(\varphi)} \tag{9}$$

$$\sigma^{T*} = R\sigma_i^T \tag{10}$$

where:

- UCS* is the new UCS value after weakening;
- $c^*$ is the new cohesion after weakening;
- $\sigma_i^T$ is the initial intact tensile strength;
- $\sigma^{T*}$ is the new tensile strength after weakening.

Equations (5)–(10) provide the basis needed to begin verifying the model at the lab scale and applying it at the excavation scale. Figure 5 provides a schematic of the weakening behaviour as described by the LTS model and Figure 6 provides a schematic workflow for the overall strength degradation model when implemented into a finite-difference modeller such as FLAC2D.

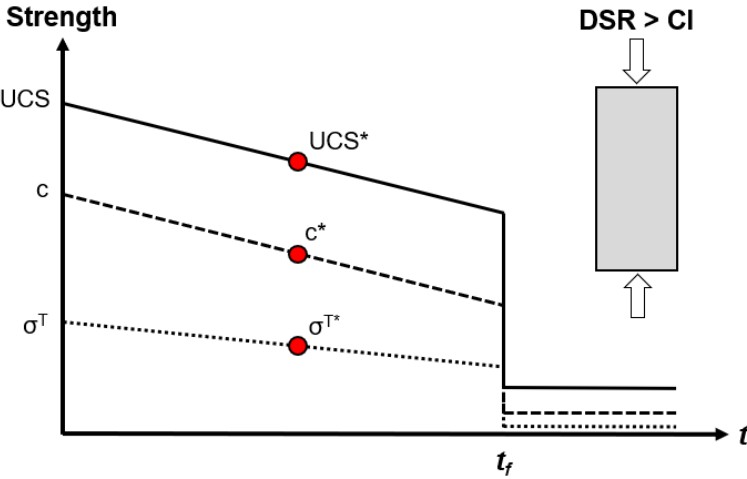

**Figure 5.** Schematic representation of the LTS model and the effect on the respective strength parameters when a sample is subjected to an applied constant load greater than its CI threshold. Note that friction angle has been omitted as it is assumed constant until failure. Values after tf are residual (post-yield), * denotes the change over time.

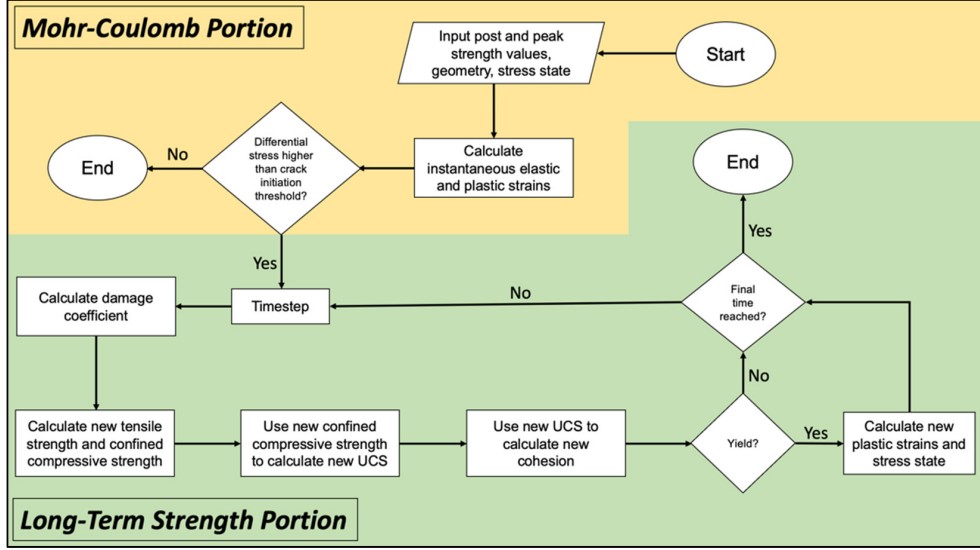

**Figure 6.** Workflow of strength-degradations portion for the long-term strength model for application into finite difference continuum models.

### 3.1. Effect on Applied Stress and Confinement on Secondary Strain-Rates and Viscosities

It has long been shown that brittle rocks do experience time-dependent strains when under applied stress, e.g., [36,48,82–89]. It is generally theorized that the strains associated to creep in brittle rocks are not due to true creep mechanics (continuum processes), but rather, crack initiation and propagation with time which are associated to brittle failure mechanisms. Continuum models (such as those created in FLAC2D) cannot explicitly capture brittle behaviour due to the discontinuum nature of brittle failure. Therefore, to capture these strains, the Burgers model for creep can be implemented in a model to account for brittle deviatoric strains (recall that Burger's creep model is deviatoric only). It is important that the time-dependent aspect of brittle failure is well understood as creep at the tunnel scale allows for stress relaxation [70].

The author of [39] studied the effect of creep in steel and showed that the secondary strain rate can be expressed as shown in Equation (11).

$$\dot{\varepsilon}^{\mathrm{s}} = \alpha \bar{\sigma}^{\beta} \tag{11}$$

$$\bar{\sigma} = \sqrt{3J_2} = \left(0.5[(\sigma_{11} - \sigma_{22}) + (\sigma_{22} - \sigma_{33})^2 + (\sigma_{33} - \sigma_{11})^2 + 6(\sigma_{12}^2 + \sigma_{23}^2 + \sigma_{31}^2)]\right)^{0.5} \tag{12}$$

where:

- $\dot{\varepsilon}^{\mathrm{s}}$ is the secondary strain rate;
- $\bar{\sigma}$ is the equivalent Von–Mises stress;
- $J_2$ is the second invariant of the deviatoric stress tensor;
- $\sigma_{\mathrm{ii}}$ and $\sigma_{\mathrm{ij}}$ are components of the Cauchy stress tensor;
- $\alpha$ and $\beta$ are curve-fitting constants.

Conversely, the secondary strain rate as described by the Maxwell and Burgers equations is shown in Equation (13).

$$\dot{\varepsilon}^{\mathrm{s}} = \frac{q}{3\eta_{\mathrm{M}}} \tag{13}$$

From Equation (11), it is clear that with a change in deviatoric stress, the secondary strain rate changes linearly, assuming that the secondary viscosity term ($\eta_{\mathrm{M}}$) is constant whereas the secondary strain rate as described by Equation (12) changes exponentially with changes in stress, as described in [70]. This means that the behaviour as described by the Power Law is for non-Newtonian fluids, whereas the behaviour as described by the rheological models is for Newtonian fluids. The key behaviour of Newtonian fluids is that the viscosity of the material does not change with stress, strain-rate, or deformation, but a non-Newtonian fluid's viscosity is not constant [60]. In practice, it is typical that one average value is used for the secondary viscosity in the Burgers/CVISC model [31,88–90]; however, as shown in Figures 7 and 8, this is only applicable when the expected stresses are monotonic and unchanging from the lab-calibrated values.

A general equation for the secondary viscosity, or Maxwell viscosity ($\eta_M$), as shown in Figure 7 can be written in Equation (14) as:

$$\eta_{\mathrm{M}} = \chi \exp(\kappa q) \tag{14}$$

where:

- $\chi$ is some function of confinement;
- $\kappa$ is the rate of change in secondary viscosity with changing deviator stress;
- $q$ is the deviator stress.

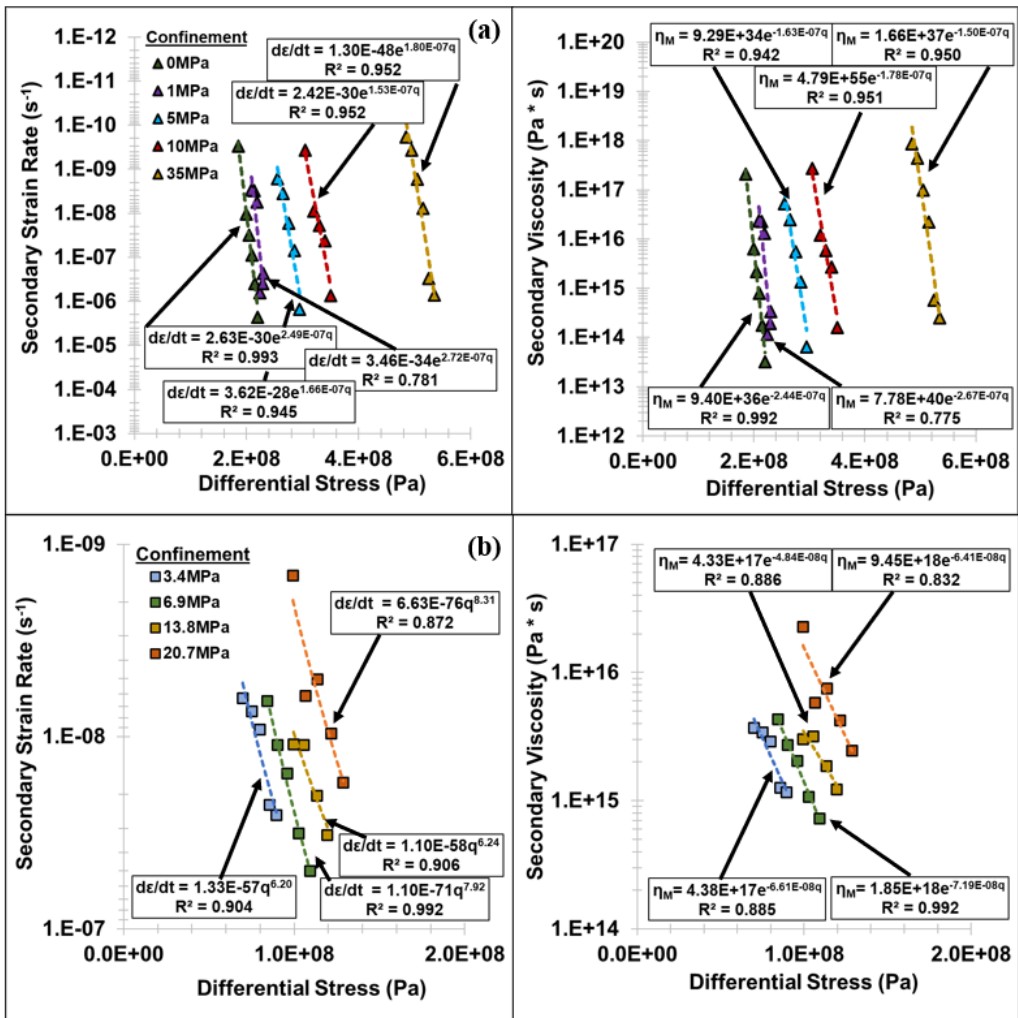

**Figure 7.** Secondary strain rates and associated Maxwell viscosities at varying levels of confinement for (**a**) LdB granite from [77], (**b**) Gyda sandstone from [91]. Note that $\eta_M$ denotes Maxwell viscosity, q is the deviatoric stress, and $d\varepsilon/dt$ is strain rate.

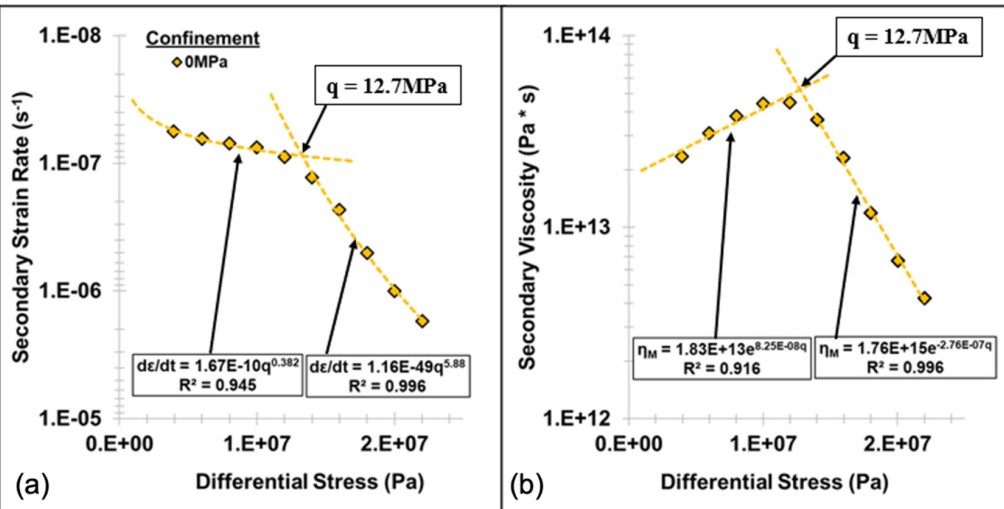

**Figure 8.** (**a**) Secondary strain rate and (**b**) Maxwell viscosity versus applied stress for rock salt from [92].

As shown in Figure 7, viscosity is also dependent on confinement; however, the rate of change ($\kappa$) in viscosity at any confinement level is constant. To effectively capture this behaviour of confinement dependency, the behaviour of the variable '$\chi$' must be determined. Figure 9 shows the change in '$\chi$' with confinement assuming an average value of $-2.004 \times 10^{-7}$ Pa$^{-1}$ for '$\kappa$' for LdB granite. From Figure 9, there is a clear pattern in '$\chi$' with confinement, with it increasing exponentially with increased confinement. Plugging in the Equation shown in Figure 9 into Equation (14), the Maxwell viscosity for LdB granite is fully expressed in Equations (15) and (16) as:

$$\eta_M = [4.28\text{E} + 34 \exp(1.77\text{E} - 6(\sigma_3))] \exp(-2.004\text{E} - 7(q)) \tag{15}$$

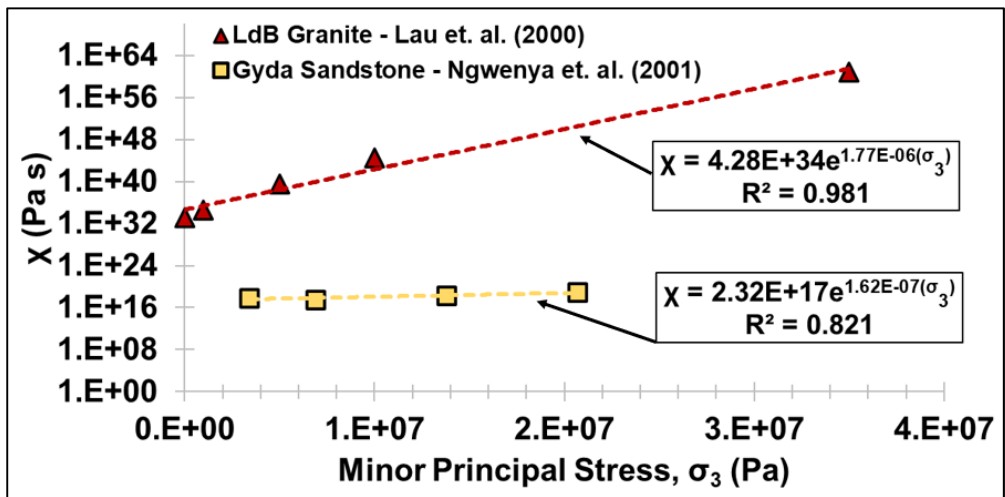

**Figure 9.** Plot of variable $\chi$ versus confinement for LdB granite from [77] and Gyda sandstone from [91].

Which simplifies to:

$$\eta_M = 4.28\text{E} + 34 \exp[1.77\text{E} - 6(\sigma_3) - 2.004\text{E} - 7(q)] \tag{16}$$

where:

- $\eta_M$ is the Maxwell (secondary) viscosity;
- q is the deviator stress ($\sigma_1 - \sigma_3$);
- $\sigma_3$ is the minor principal stress.

The resulting viscosities for LdB granite in the AECL URL tunnel are shown in Figure 10. Note that this is an empirical approach to accounting for effects of confinement in brittle rock creep and that none of the constants in Equation (16) have any significance to real-world mechanisms. Additionally, it has been shown that a Maxwell material most likely behaves as a non-Newtonian fluid, rather than as a Newtonian fluid, and the same can likely be said for a Kelvin type material; however, this distinction is out of the scope of this analysis. In simple loading conditions, such as those shown in the following numerical models, strains are more sensitive to secondary viscosities over long periods of time rather than primary viscosities; therefore, the distinction between Newtonian and non-Newtonian viscosity for primary creep is insignificant.

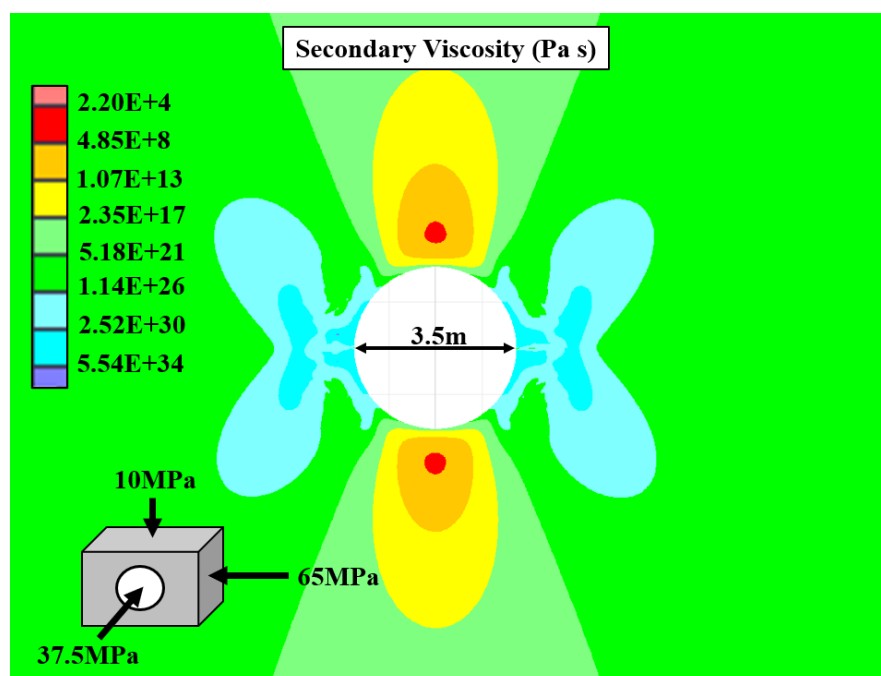

**Figure 10.** Instantaneous secondary viscosity contours around the 3.5 m diameter AECL URL tunnel in LdB granite.

### 3.2. Lab Scale Verification of Model

The long-term strength and modified CVISC model are implemented into FLAC2D [65], which is a finite-difference continuum numerical modelling software. This section will provide a lab-scale verification of the model as well as a tunnel-scale analysis using the AECL URL geometry in LdB granite. The material properties used are shown in Table 3, which are average values for LdB granite. The strength values are from [8], and the creep properties are from [19], which were calculated using lab data from [77]. The goal of the lab-scale verification model is to show that under varying confinement conditions, the proposed long-term strength model accurately captures the predicted analytical time-to-failure.

**Table 3.** Intact strength, long-term strength, and creep parameters for LdB granite. Peak and residual strength values calibrated from [8] and time-dependent values from [19,77]. Variables K, G, c, φ, $\sigma^T$, and η denote bulk modulus, shear modulus, cohesion, friction angle, tensile strength, and viscosity, respectively. Additionally, subscripts p, r, M, and K denote peak, residual, Maxwell, and Kelvin, respectively. Variables A, B, and C are shown in Equation (2). Note that 'E' is scientific notation (i.e., 4.08E + 14 = $4.08 \times 10^{14}$) and 'exp' denotes and exponential of '*e*' (i.e., exp(x) = $e^x$).

| Parameter | Value |
|---|---|
| K (GPa) | 58 |
| G (GPa) | 25 |
| $c_P$ (MPa) | 40 |
| $c_r$ (MPa) | 0.1 |
| $\varphi_P$ (deg) | 50 |
| $\varphi_r$ (deg) | 22 |
| $\sigma^T_P$ (MPa) | 8 |
| $\sigma^T_r$ (MPa) | 0 |
| $\eta_M$ (Pa s) | $4.28\mathrm{E}+34\exp[1.77\mathrm{E}-6(\sigma_3)-2.004\mathrm{E}-7(q)]$ |
| $\eta_K$ (Pa s) | 4.08E + 14 |
| $G_K$ (GPa) | 107 |
| A | 1.18 |
| B | 0.084 |
| C | 3.81 |

The geometries used for both the lab-scale verification and excavation scale models are shown in Figure 11 with the FLAC grid. The grid for the lab verification is relatively coarse for computing efficiency and should not affect the TTF in any significant way. Additionally, the lab-scale model uses a DSR of 0.75 for each confinement level, which is achieved by changing the applied stress on the top and bottom of the sample. This is performed in such a way that the TTF is the same for each run and making comparisons between them is easier. The excavation scale model uses stresses modified from the AECL URL tunnel as provided in [8]. The most notable change is the change in out-of-plane stress which was changed from the published 43 MPa to the 37.5 MPa (average of in-plane stresses such that out-of-plane stress is effectively ignored). The grid in the tunnel model is radial with a very dense mesh around the excavation, becoming gradually coarser towards the model boundaries, which are 17.5 m from the tunnel centre.

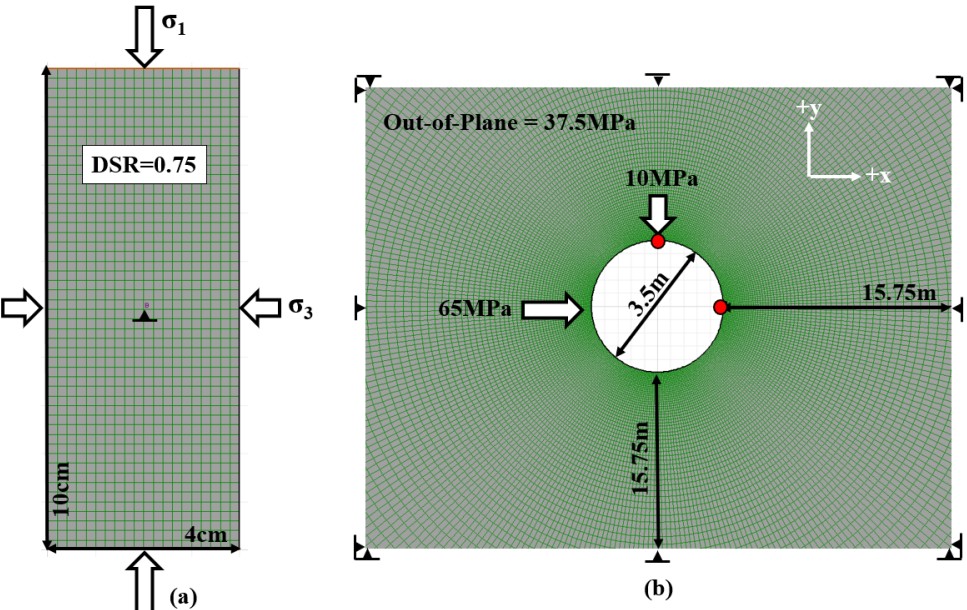

**Figure 11.** Model geometries from FLAC2D for (**a**) the lab-scale verification and (**b**) the tunnel-scale model modified from the AECL URL tunnel. Note that model dimensions and boundary conditions (pins) are shown. Red dots represent measurement points for displacement.

The results of the lab-scale verification can be seen in Figure 12, which plots the axial strain and cohesion of the sample versus time with the analytical TTF overlain as a dotted red line. The effective secondary viscosity is also shown and is calculated from Equation (16); however, each of the values is high enough that it is insignificant in the time span used. Failure in each of the models is represented by the marked rapid decrease in cohesion to its residual state (0.1 MPa) followed by the rapid increase in strain rate. The intact cohesion value for each confinement level at failure is shown to decrease with increasing confinement as expected. The lag between cohesion loss and strain increase can be attributed to the increase in unbalanced forces resulting from failure. From each of the plots in Figure 12, it is presented that the loss in cohesion correlates with the analytical TTF, showing that, as formulated, the long-term strength model is adequate and can be carried over to a tunnel-scale model.

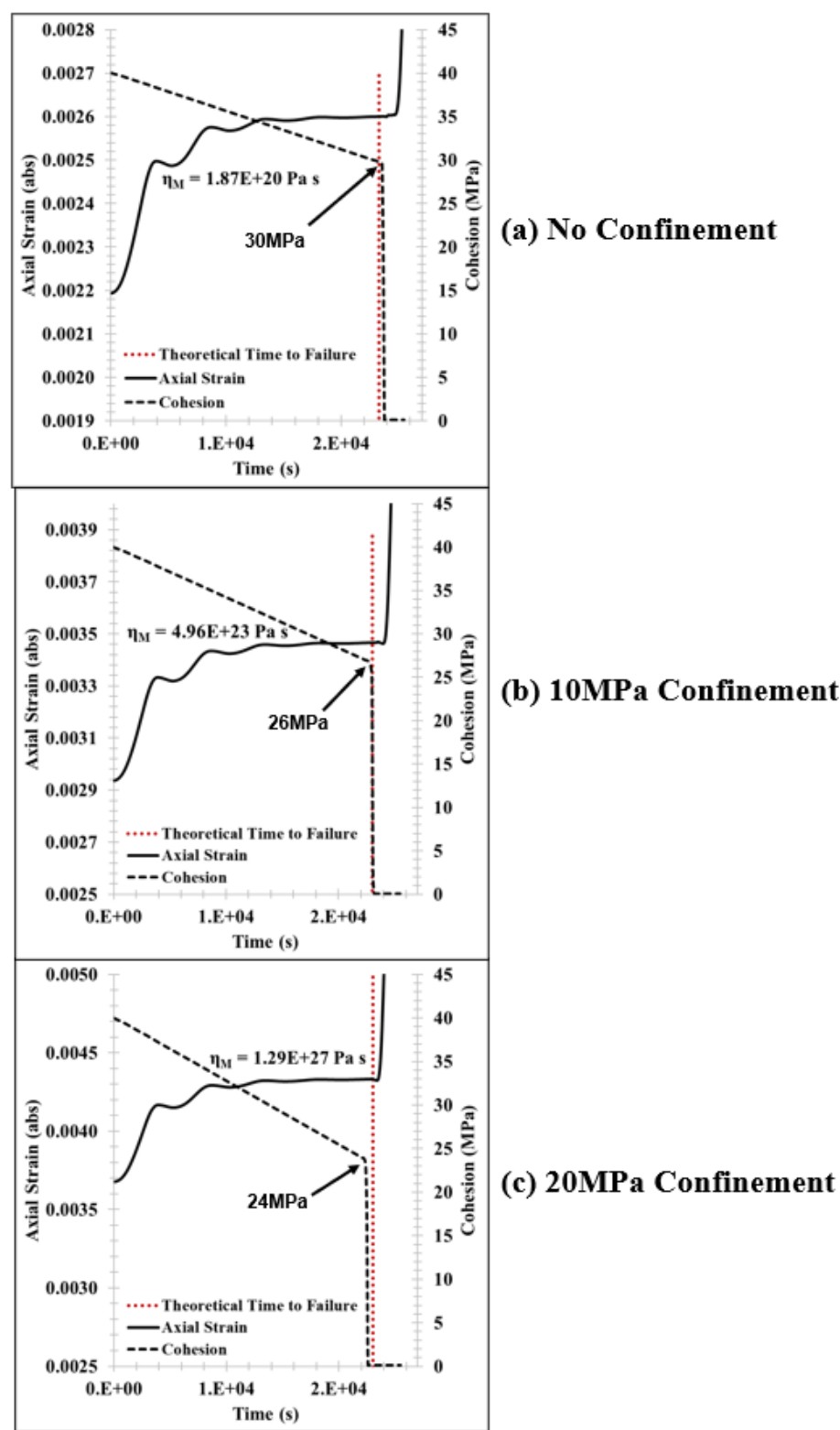

**Figure 12.** Axial strain and cohesion with time under a constant applied stress for (**a**) no confinement, (**b**) 10 MPa confinement and (**c**) 20 MPa confinement. All models are run with a DSR of 0.75 and failure is marked by the rapid decrease in cohesion with time. The cohesion value at failure is shown for each run.

### 3.3. Tunnel Scale Model

The tunnel-scale model for LTS is developed using the geometry shown in Figure 11. This geometry is similar to that shown in [8] for the AECL URL. In the model, the tunnel is developed "instantaneously" in a Mohr–Coulomb strain-softening medium using the values as shown in Table 3. At the tunnel scale, the goal of the LTS model is to accurately capture the typical tension cracks that form in the walls of the tunnel parallel to the minimum principal stress as well as the brittle overbreak "notch", described by [7,8,10,76], that forms in the periphery of the tunnel in the direction of the maximum applied stress. To compare and validate the applicability of the LTS model, it will be compared to a CWFS model run in the exact same conditions using the values shown in Table 4.

**Table 4.** Values used for the CWFS validation modified from [8]. Note that $\varepsilon_c^P$ and $\varepsilon_\varphi^P$ are plastic strain values for residual state to be reached for cohesion and friction angle, respectively.

| Parameter | Value |
|:---:|:---:|
| K (GPa) | 58 |
| G (GPa) | 25 |
| $c_P$ (MPa) | 40 |
| $c_r$ (MPa) | 0.1 |
| $\varphi_P$ (deg) | 20 |
| $\varphi_r$ (deg) | 50 |
| $\sigma^T_P$ (MPa) | 8 |
| $\sigma^T_r$ (MPa) | 0 |
| $\varepsilon_c^P$ (%) | 0.3 |
| $\varepsilon_\varphi^P$ (%) | 0.3 |

The in-situ stresses and associated strains from the CWFS model are shown in Figure 13 and the instantaneous response from the LTS model is shown in Figure 14. From comparing the results in Figures 13 and 14 (left), it can be seen that the typical Mohr–Coulomb-based analysis is not adequate for capturing the brittle overbreak that is observed in the CWFS analysis as no failure occurs instantaneously according to the LTS model. Figure 14 (right) then shows the in-situ stresses and strains from the LTS model after 7 h, the time of yield in the floor and roof of the tunnel. Here, the stresses have relaxed around the roof and the floor and redistributed accordingly, indicating rupture. The shear and volumetric strains do not, however, match the pattern as shown in the CWFS analysis and can likely be attributed to numerical noise.

To validate the results of the LTS model, the failure geometries must be compared. From the CWFS model, a typical "notch" forms in the roof and the floor to a depth of 0.75 m to 0.80 m as well as tension cracks forming in the walls. From the LTS model, the tension cracks in the wall form instantaneously, but the degree of failure in the walls increases at 7 h. The degree of wall displacement also matches the displacement as shown in the CWFS model. The failure in the roof and the floor of the LTS model also shows somewhat of a notch-type geometry to a depth of 0.70 m, which is marginally less than the depth as predicted by the CWFS model. When looking at the DSR in Figure 14, it can be seen that, at the depth of failure, deviatoric stresses are at about 70% of the strength of rock, indicating that further yield may occur with more time. The final displacements in the roof after failure in both models also match within marginal error.

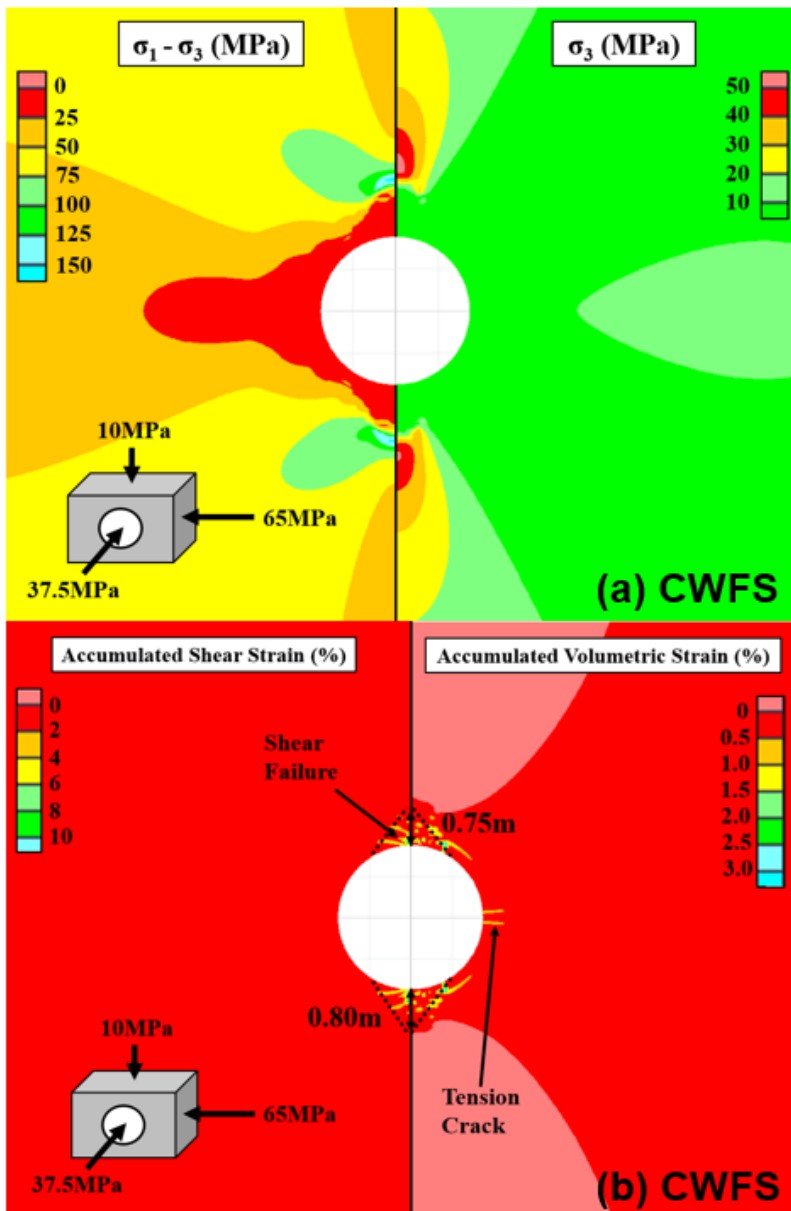

**Figure 13.** Instantaneous response from the CWFS analysis showing (**a**) in–situ stresses and (**b**) shear and volumetric strain with depth of yield and typical "notch"-type failure.

From Figures 13 and 14, it is clear that the failure modes between the CWFS and the LTS models differ, wherein the CWFS model, multiple shear bands develop parallel to the tunnel periphery and in the LTS model, a shear "cone" extends outwards from the tunnel periphery. The failure geometry shown by the LTS model is similar to what some researchers refer to as the "process zone" [22,23,76]. The term "process zone" refers to the small-scale buckling that occurs at the tip of the notch-type failure that gives rise to more considerable dilation, as observed in the CWFS model. Therefore, if the LTS model were to be run for even more time after the initial nucleation of the process zone, the failure geometry may begin to resemble that as shown in the CWFS model. The process zone development is described in [22] and the observed development within the AECL URL is shown in Figure 15.

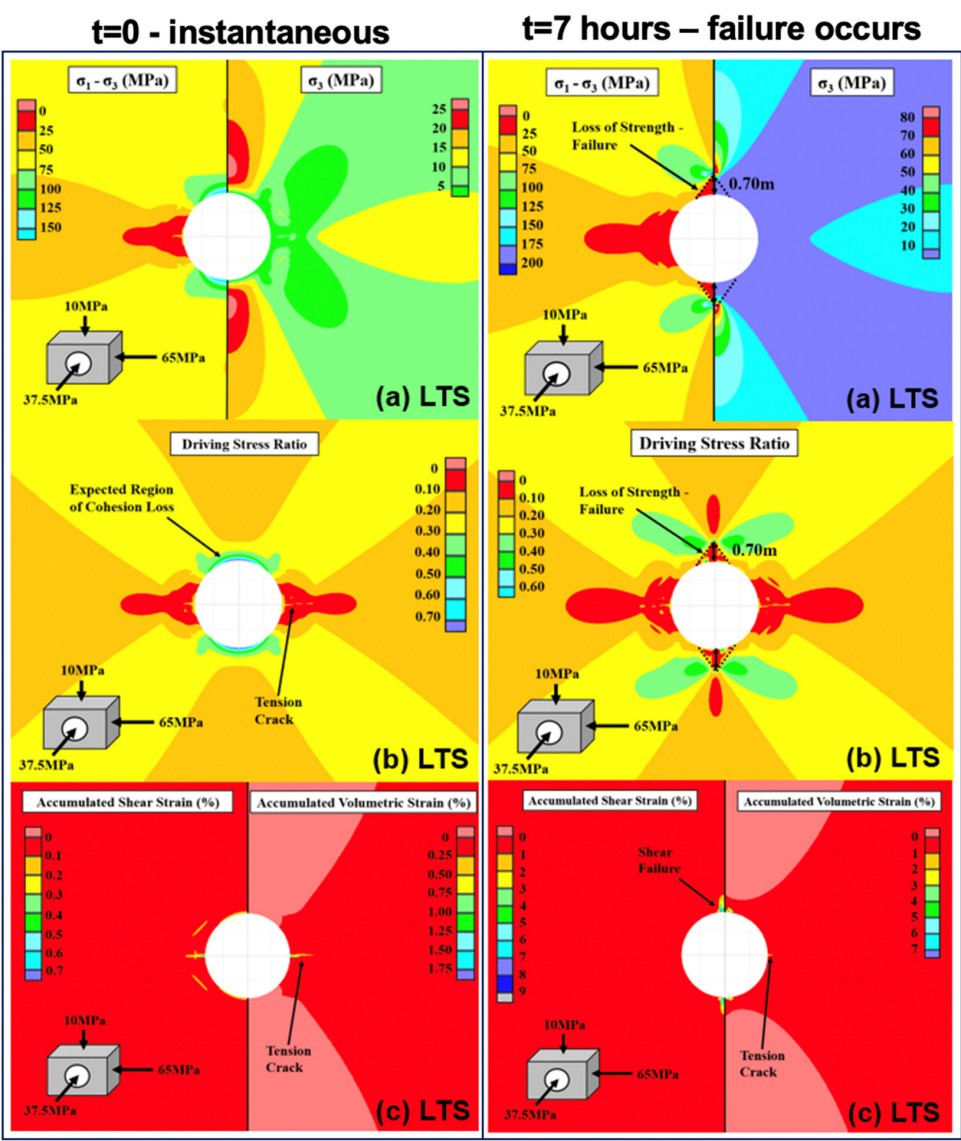

**Figure 14.** Instantaneous using plasticity. Left: (**a**) in–situ stresses, (**b**) driving stress ratio, and (**c**) shear and volumetric strains and after 7 h (failure time) using the LTS model. Right: (**a**) in-situ stresses, (**b**) driving stress ratio with depth of failure, and (**c**) shear and volumetric strains for the LTS model.

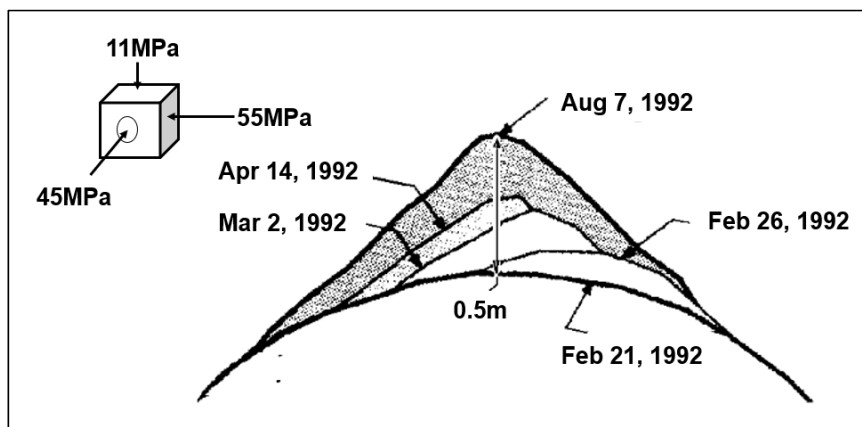

**Figure 15.** Observed yield profile development within the AECL URL from [22].

## 4. Discussion

The effect of changing deviator stress and confinement is often not considered when determining parameters for the Burgers creep model. From Figures 7 and 8, it is shown that using a single value for viscosity as shown in the Burgers model is not adequate when describing both brittle and ductile creep processes when complex loading conditions are expected. The Bailey–Norton two-component power law addresses both the exponential change in viscosity with stress as well as the transition from ductile to brittle creep at some stress. The effect of confinement can be addressed using the curve-fitting technique used in Figure 9. Using the curve-fitting technique proposed to determine secondary viscosity as a function of confinement, the viscosities throughout a tunnel model can be calculated as shown in Figure 10. The implications of using viscosities that are too low in a tunnel model, as is often the scenario when using average values from lab testing, is that tunnel convergence becomes very large compared to the observed values as well as allowing for too much stress relaxation around the tunnel periphery.

The failure process as demonstrated in Figure 13 according the LTS model assumes that up until material rupture, the rock stays intact throughout the damaging process. This contrasts with the discontinuum models demonstrated in [36,79] which showed that in discontinuum models, cracks do initiate and accumulate with time under a constant load. This highlights the advantage of using discontinuum models over continuum models when modelling brittle failure processes, which are inherently discontinuous. In contrast, [49] showed that using the Burgers creep model to model tunnel convergence along a longitudinal profile in a continuum setting in creeping ground is adequate because creep is inherently a continuum process.

The LTS model has successfully been implemented into a continuum finite-difference model to both model unconfined and heavily confined cylindrical samples of rock at the lab scale and, when modelling, the brittle overbreak encountered high deviatoric stresses at the tunnel scale. At confinement levels of 0, 10, and 20 MPa at the lab scale and DSRs of 0.75, the sample yields as denoted by a rapid increase in strain rate, similar to that of tertiary creep from Figure 2. It is important to note that the LTS model does not consider visco-plasticity unlike the simplified Cividini and Gioda model. At the tunnel scale, the LTS model must be validated to other established numerical models used to simulate brittle failure, such as the DISL and CWFS models. Using the geometry and stress conditions similar to those encountered in the AECL URL in LdB granite, both the CWFS and LTS models show similar convergence measurements but differing yield shapes after first failure as shown in Figures 13 and 14. The first failure as described by the LTS model is similar to what is referred to as the process zone, which is the preceding mechanism to spalling.

The advantage of the LTS model over other continuum-based brittle failure models is that the time it takes for yield to occur can also be calculated in addition to yield geometries. This provides further guidelines for engineering design in terms of timelines for installation support. Allowing engineers and researchers to predict TTF at the excavation scale can lead to project and support optimizations in brittle rocks, reduction of uncertainties, e.g., [93–98] in the design while avoiding tunnel failures [99] and reduction or even overcoming of cost overruns, e.g., [100–103]. This model can be modified to calculate stresses and strains in three dimensions as well as for analysis along a longitudinal displacement profile for further tunnelling optimizations.

In addition, the LTS model is relatively simple to implement and obtain parameters for, whereas more complex visco-plastic models require multiple inputs and complex lab testing to obtain parameters for and become less feasible for an engineering analysis. The workflow of the LTS model is shown in Figure 6 and the associated rheological analogue is shown in Figure 16, highlighting the necessary modifications to the existing CVISC model.

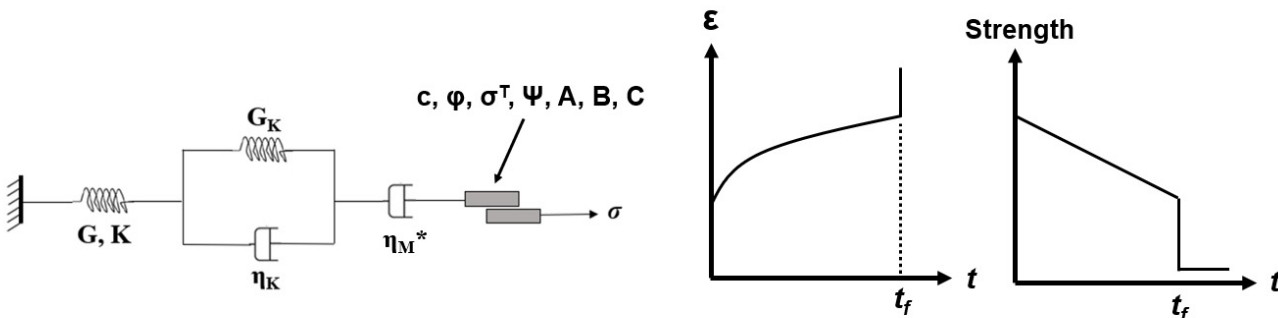

**Figure 16.** Conceptual schematic of the LTS model.

## 5. Conclusions

In this paper, a proposed long-term strength (LTS) model is developed for use in continuum models based on time-to-failure data for brittle rocks exploration at the lab scale. The advantage of this model over the conventional CWFS and DISL models is that, in addition to being able to simulate brittle breakout around tunnel peripheries, it can also simulate the time for such failure to occur, providing more information for engineering analysis and design. The model is a modified version of the CVISC creep model, where the secondary viscosity is modified to act as a non-Newtonian fluid and the Mohr–Coulomb slider experiences cohesive degradation based on the ratio of in-situ stress to strength, or driving-stress ratio, based on empirical relationships. Cohesion loss can be associated to the initiation and propagation of cracks through the material, reducing the overall effective cohesion. This opening and propagating of fractures also reduces tensile strength, which is considered as well. The friction angle is assumed constant until a residual state is reached.

The LTS model is built upon TTF lab testing wherein a cylindrical sample of rock is subject to a constant stress greater than its respective CI and less than instantaneous strength and is used to predict the ultimate stability time for any brittle rock subject to some deviator stress. The current state of practice for accounting for strength loss with time in a numerical model is to manually decrease strength parameters of the rock in stages, which results in bulk weakening; however, as shown in this analysis, only areas of rock subject to high deviator stresses weaken due to stress corrosion. The LTS model provides several advantages over classical numerical modelling techniques, including CWFS analyses, allowing engineers and scientists to weaken specific areas of rock with time using lab data as a basis. As such, a more precise analysis on the timing for support installation and excavation step sizes can be made. The model in its current state calculates TTF based solely on unconfined TTF lab tests, which likely does not reflect real-world excavation scale behaviour. Finally, it should be highlighted that to calibrate the model, further research is needed on the effects of confinement on TTF as well as field scale convergence measurements in longer time periods to ensure model validity.

**Author Contributions:** Conceptualization, J.I., C.P. and M.S.D.; methodology, J.I., C.P. and M.S.D.; software, J.I.; validation, J.I.; formal analysis, J.I.; investigation, J.I. and C.P.; resources, M.S.D.; data curation, J.I. and C.P.; writing—original draft preparation, J.I.; writing—review and editing, C.P. and M.S.D.; visualization, J.I., C.P. and M.S.D.; supervision, C.P. and M.S.D.; project administration, C.P.; funding acquisition, M.S.D. All authors have read and agreed to the published version of the manuscript.

**Funding:** This research was funded by Nuclear Waste Management Ontario—NWMO, grant number GS830 and Natural Sciences and Engineering Research Council of Canada—NSERC CRDPJ 523562-18.

**Data Availability Statement:** Not applicable.

**Acknowledgments:** The authors would like to acknowledge the contributions made by the Nuclear Waste Management Organization (NWMO) and the Natural Sciences and Engineering Research Council of Canada (NSERC) for this research would not have been possible without their continued support.

**Conflicts of Interest:** The authors declare no conflict of interest.

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
