# Peer review of "Time-Dependent Model for Brittle Rocks Considering the Long-Term Strength Determined from Lab Data"

_mining, doi:10.3390/mining2030025_

Round 1

Reviewer 1 Report

The paper uses a time-dependent continuum constitutive model (LTS) to predict time to failure of a simulated laboratory sample of granite. The same model is then used in a numerical simulation of the URL tunnel, and the displacements and failure modes compared both to the field observations and the response using a non-time dependent constitutive model (CWFS).

Simulations of time-dependent behavior of engineered structures in hard, brittle rock are hardly every performed. (For example, an otherwise-comprehensive textbook (“Guidelines for Open-Pit Slope Design”) does not even list “time-dependence” in the index, although many examples are given of ways to measure the evolution of displacement with time). Thus, the aim of the authors of the present paper is to be encouraged. The paper is very well written and the literature extensively cited, but there are unfortunate shortcomings that detract from its merit. First, the tunnel simulation is only performed for an elapsed time of 7 hours, while the yielding and spalling at the URL took place over 5 months, which is 500 times the period of the simulation. Second, the time histories of strain shown in Figure 12 show oscillations, rather than the monotonic evolution that would be expected. These seem to indicate that the unbalanced forces in the simulations of the lab scale validation were allowed to be too high, thus causing dynamic effects to become evident. There is guidance in the FLAC manual concerning the way to limit timestep changes to keep the maximum unbalanced force below a certain threshold. The run time should not be a problem for a small sample of 1000 zones, although it is understood that run time may be a challenge for the tunnel model.

Despite the shortcomings of the simulations, the paper is a good first step in suggesting a means to account for time-dependent stress-corrosion in hard, brittle rock. I avoided recommending “accept with revisions” because I believe that any such revisions would necessitate completely re-working the simulations. However, if the authors wish to do that, a better paper may be the result.

Author Response

The authors would like to thank for his/her valuable comments. We take into considerations these and in future numerical simulations will try to implement this. We believe for the purpose of this paper we would like to show the bigger picture on how a model can exist that can take into consideration this time-dependent behaviour in hard brittle materials. The authors are encouraged to continue research on this field to optimise the simulations. The authors added this sentence to highlight the importance of further research.

“Finally, it should be highlighted that to calibrate the model, further research is needed on the effects of confinement on TTF as well field scale convergence measurements in longer time-periods to ensure model validity.”

Reviewer 2 Report

The paper introduced and implemented the Long-Term Strength (LTS) model into FLAC2D. The LTS model is verified against its corresponding analytical solution using a constant stress creep lab test and implemented into a tunnel scale model using the geometry, stress, and geologic conditions from the Atomic Energy of Canada Limited Underground Research Laboratory (AECL URL). It is a research work of great theoretical value and engineering significance.

Comments 1: The paper uses too many words to introduce the research background, it is suggested to simplify it.

Comments 2: How many specimens are used for testing? If there are too few experimental specimens and data, the accuracy and representativeness of the model cannot be guaranteed.

Comments 3: Are the numerical simulation conclusions verified and supported by corresponding engineering case phenomena? In this way, the applicability and accuracy of the model can be better illustrated.

Comments 4: It is suggested that major adjustments and modifications should be made to the conclusion of this paper. It is only necessary to list the important conclusions of this paper one by one, without summarizing the research background and research methods.

Author Response

The authors would like to thank for his/her valuable comments and suggestions. The authors took these into considerations and made the appropriate changes where applicable.

Comments 1: The paper uses too many words to introduce the research background, it is suggested to simplify it.

The authors would like to thank for his/her valuable comments, simplifications were made in the text. However, the authors believe that the background is important for developing a better understanding to readers who are new to this field (time-dependent behaviour in brittle materials).

Comments 2: How many specimens are used for testing? If there are too few experimental specimens and data, the accuracy and representativeness of the model cannot be guaranteed.

The authors would like to thank for his/her valuable comment. The dataset is derived from laboratory testing already performed by other researchers that have published their data. The lab tests that this research is based on are more than 150. This research work is the continuation of the following paper published in the International Journal of Rock Mechanics and Mining Sciences.

Innocente JC, Paraskevopoulou C, Diederichs MS. Estimating the long-term strength and time-to-failure of brittle rocks from laboratory testing. Int J Rock Mech Min Sci [Internet]. 2021;147. Available from: https://doi.org/10.1016/j.ijrmms.2021.104900

The following sentence was added to the text.

“Using Equation (4), the strength of the rock at the excavation scale can be determined throughout the FLAC2D [65] grid. With the data from Figure 3, the theoretical time-to-failure can be determined throughout the FLAC2D grid using the Equations described below. A key assumption in the development of this model is that the failure resulting from stress corrosion is the result of a linear loss in cohesion with time that is a function of the in-situ stress conditions. This assumption is made based on the log-linear relation that laboratory data shows when brittle rock materials are subjected under constant load as [19] thoroughly analysed and discussed lab datasets from a range of rock materials”

Comments 3: Are the numerical simulation conclusions verified and supported by corresponding engineering case phenomena? In this way, the applicability and accuracy of the model can be better illustrated.

The authors would like to thank for his/her valuable comment. The model is verified based on the log-linear relationship that us derived from the experimental data from the previous paper:

 Innocente JC, Paraskevopoulou C, Diederichs MS. Estimating the long-term strength and time-to-failure of brittle rocks from laboratory testing. Int J Rock Mech Min Sci [Internet]. 2021;147. Available from: https://doi.org/10.1016/j.ijrmms.2021.104900

The model is then used to simulate a real engineering case study the AECL lab.

Comments 4: It is suggested that major adjustments and modifications should be made to the conclusion of this paper. It is only necessary to list the important conclusions of this paper one by one, without summarizing the research background and research methods.

The authors would like to thank for his/her valuable comment. The conclusions have been modified as suggested.
